# Use of a Gamified Platform to Improve Scientific Writing in Engineering Students

Rosa Núñez-Pacheco [1,*], Elizabeth Vidal [2], Eveling Castro-Gutierrez [2], Osbaldo Turpo-Gebera [3], Aymé Barreda-Parra [4] and Ignacio Aguaded [5]

1   Departamento Académico de Literatura y Lingüística, Universidad Nacional de San Agustín de Arequipa, Arequipa 04000, Peru
2   Departamento Académico de Ingeniería de Sistemas e Informática, Universidad Nacional de San Agustín de Arequipa, Arequipa 04000, Peru; evidald@unsa.edu.pe (E.V.); ecastro@unsa.edu.pe (E.C.-G.)
3   Departamento Académico de Educación, Universidad Nacional de San Agustín de Arequipa, Arequipa 04000, Peru; oturpo@unsa.edu.pe
4   Departamento Académico de Psicología, Universidad Nacional de San Agustín de Arequipa, Arequipa 04000, Peru; vbarredapa@unsa.edu.pe
5   Departamento de Educación, Universidad de Huelva, 21004 Huelva, Spain; aguaded@uhu.es
*   Correspondence: rnunezp@unsa.edu.pe

**Abstract:** The general purpose of this study was to determine the potential of using a gamified platform in the development of scientific writing skills among engineering students at a Peruvian university. To this end, a gamified web platform named *Call for Papers for Engineers* was designed. This platform contains mini-games focused on developing reading and writing skills for articles related to the engineering area. A quantitative methodological approach was employed, with a quasi-experimental design involving two groups: an experimental group and a control group, with pre-and post-test measurements. Additionally, the gamified platform was validated through expert judgment, and user satisfaction levels were assessed. The main results indicate that the content developed in the course and the use of the gamified web platform were effective teaching methods, as the students in the experimental group demonstrated higher performance after using the gamified platform compared to the control group. Furthermore, participants in the study expressed satisfaction with the use of this technological resource, finding it motivating and user-friendly.

**Keywords:** gamified platform; gamification; scientific writing; engineering; motivation





## 1. Introduction

Various methodologies have contributed to improving students' writing competence, such as project-based learning [1], problem-based learning [2], blended learning, and gamification [3], among others. These methodologies have enhanced various educational contexts by promoting active and collaborative participation in teaching and learning processes. Gamification has gained traction in different educational spaces only in recent years.

Traditionally, gamification has been defined as the application of game elements in non-game activities [4,5], with the primary goal of enhancing participants' intrinsic motivation. Several authors have emphasized different aspects of gamification. Ref. [6] highlight the importance of evoking psychological experiences like those generated by games through gamified processes. On the other hand, ref. [4] emphasizes the implementation of game elements in the gamified process, regardless of the outcomes achieved [7].

In higher education, gamification has emerged as a relevant tool to motivate university students in content development and classroom participation [8,9]. It involves the use of game mechanics in educational environments, providing an opportunity to work on aspects such as motivation, effort, engagement, and cooperation, among others. Gamification motivates and establishes a connection between the student and the content being

studied, changing their perspective, and enabling better absorption of knowledge, skill improvement, or rewarding specific actions [10]. To implement gamification in learning, it is necessary to transform educational materials, adapting them to the experiences and expressive forms of the digital society.

In higher education, gamification has shown positive results; however, it is important to investigate whether these same results can be extrapolated to the entire university population. Bicen and Kocakoyun [11] point out that the effects of gamification can vary among different participants, including school students, university students, doctoral candidates, and others, highlighting the need to examine its effects on different samples. Additionally, it is worth considering that although students are considered "digital natives", some teachers may not fully appreciate the use of games as a means of learning.

Despite the abundance of literature on gamification in higher education, there are not many specific studies focused on its application in science and engineering education [12]. This suggests that, although the concept of gamification generates significant interest and has driven research efforts, there is still a lack of a solid theoretical framework that provides a satisfactory explanation of this phenomenon. One of the contributing factors to this lack of clarity is the scarcity of shared definitions and a common taxonomy for classifying this concept, leading to ambiguities in the terminology used [13].

In particular, the use of playful resources in engineering education goes beyond acquiring knowledge and developing skills. Games have significant potential to motivate engineering students and drive innovation projects [14,15]. University students report having gained competencies in areas such as quality, creativity, and problem-solving [16]. They also indicate an improved understanding of the efforts and tensions associated with each of these competencies, which is considered a specific skill. In terms of emotions, students primarily report experiencing positive emotions, while negative emotions have been less frequent. Regarding the workload, students do not feel overwhelmed and consider the time spent on the activity to be similar to what was initially planned [17].

Likewise, the use of gamification in higher education has favored the development of communicative competence, especially writing [18,19] (El Tantawi et al., 2018; Gallego and Agredo, 2016). As writing is a necessary skill in the education of university students, particularly in engineering, there is a need to propose new educational resources and methodological strategies that contribute to achieving this goal. In this regard, this article discusses the application of gamified strategies in the course of Writing and Communication and the use of a gamified web platform called *Call for Papers for Engineers*, primarily designed to promote scientific writing among engineering students. The overall objective of this study was to determine the potential of using a gamified platform in the development of scientific writing. To address this purpose, two specific objectives were established: (a) to analyze the level of development of scientific writing skills, and (b) to analyze the satisfaction level regarding the use of the gamified platform.

## 2. Literature Review

### 2.1. Gamification and Writing

Gamification has been used to innovate the teaching and learning process of communicative skills, particularly writing. Learning to write is considered a complex and fundamental skill for effective communication. Studies have shown that gamification enhances communicative competencies, opening up new opportunities for innovation in teaching processes [3,19]. By gamifying writing, the aim is to promote a love for reading and literary creation in students, utilizing game techniques [20], while also motivating them to venture into the world of academic writing [18]. Gamification in academic writing relates to three key aspects: game dynamics, game mechanics, and components [21].

Academic writing is a constant activity in university and post-university life, challenging students to renew, revise, and enrich their thinking. This activity encompasses writing research reports, responses to questions or problems, exposition-based debates or argumentation, and synthesis of research on specific topics [22–24]. Academic writing is

present in all disciplines and is based on consulting, reviewing, and analyzing bibliographic sources [25].

During the academic writing process, students face difficulties related to grammar, language rules, citation structuring, and textual logic. These difficulties can arise from the challenge of writing with the reader in mind, underutilizing epistemic potential, limited revision to local texts, and procrastination in writing [26]. It is necessary to consider the factors that impact the teaching of academic writing. Some studies [27–29] explain that self-perception of knowledge and mastery of writing skills have a significant impact on achieving writing competence. An objective and accurate perception of the learning process promotes greater knowledge of writing practices, which implies greater commitment and a search for solutions [30].

### 2.2. Scientific Writing in Engineering

Writing in the field of engineering involves conveying specialized information to implement solutions for practical purposes. Engineering writing has distinct characteristics that serve intermediate argumentative functions, especially useful for emphasizing and clarifying the text, such as formulas, tables, codes, algorithms, and designs, often presented as figures.

Writing is an important skill that students must develop as they graduate, as emphasized by the accrediting body ABET [31]. In addition to ABET, there is a growing demand in the industry for trained technical writers. However, for many engineering students and even professionals, the act of writing can be intimidating. Steiner's work [32] identified six points related to how engineers structure their writing tasks: (1) engineers do not receive an adequate level of composition instruction in college; (2) engineers tend to require a quiet and distraction-free space for writing; (3) outlining is the universal method used for planning writing tasks; (4) engineers value peer review highly; (5) engineers are comfortable with deadlines, and (6) engineers find deadlines very useful for prioritizing their work.

Concerns about teaching writing in engineering have been present for many years. Li [33] summarizes the main considerations for the structure of an article published in engineering journals. Emphasis is placed on making an article clear, concise, and conveying ideas in a limited space. It is also stressed that engineering results often generate numerical data, which should be presented in the form of figures or tables to facilitate reading. Pierson and Pierson [34] focus on the writing strategy of sections. The authors suggest starting with the central sections: methods and results. Within engineering, specialized scientific writing can be identified. Shaw [35] states that in software engineering, problems of various types are solved, producing various types of results that generate different types of validation evidence: procedures, techniques, qualitative or descriptive models, empirical models, analytical models, specific solutions, prototypes, software, and applications. Shaw's findings are common in many other engineering disciplines.

The concern to improve written communication skills among engineering students remains relevant. For example, Selwyn and Renaud-Assemat [36] focused on developing technical writing skills in first and second-year students. The authors highlight the importance of clearly communicating text expectations. Students received guidelines on expected style, length, structure, and content. They were also directed to other useful resources in areas such as references and writing style.

Another study by Becker and Sloan [37] integrated a technical communication instruction block into a civil engineering software applications course. The authors claim that students achieved improved technical writing competencies, including the creation of audience-focused, accessible, and usable deliverables. They also emphasize the benefits of integrating technical communication instruction, even on a small scale. Wright et al. [38] present a method to reinforce technical writing skills for mechanical engineering students. The strategies focus on the laboratory report writing process through detailed rubrics.

There are curricular experiences that incorporate engineering writing courses; however, there are many aspects that need improvement. The study by Halim et al. [39] presents the main findings: students value having more hours of writing practice in class, but they find that the content of the writing assignments is not relevant and does not relate to real engineering practice. Zemliansky and Berry [40] present the experience of designing and evaluating the effectiveness of a writing program integrated throughout the curriculum. The authors explain two implementation models: (1) direct instruction, which uses writing specialists to provide instruction to engineering and science students, and (2) the department-centered model, which instructs engineering and scientific faculty to teach writing as part of technical courses.

## 3. Materials and Methods

### 3.1. Study Design

The study has a quantitative approach, with a quasi-experimental design with two experimental and control groups applying pre- and post-test measurements.

### 3.2. Participants

The study was carried out with second year students at the professional school of Civil Engineering of a Peruvian university during the first semester of 2023. Initially, 64 students from sections A and B participated in the pre-test, which functioned as experimental and control groups, respectively. The participants were selected intentionally, based on the criterion of being enrolled in the Writing and Communication course. The post-test was administered after the semester concluded. Three students from the experimental group and one student from the control group did not respond to the questionnaire, so the final sample consisted of 60 students, with 33 in the experimental group and 27 in the control group.

The experimental group was exposed to the intervention (the use of the gamified platform *Call for Papers for Engineers* and gamification of the course), while the conventional methodology was used in the control group.

Participation was voluntary in both groups, and informed consent was obtained from all participants.

### 3.3. Instrument to Assess Knowledge

To assess their knowledge of scientific writing, a questionnaire was developed, consisting of 10 multiple-choice questions related to the preparation of a scientific article. These questions focused on topics such as choosing the most appropriate title, identifying elements of an abstract, writing citations and references, selecting keywords, composing the methodology, presenting results, discussing, and drawing conclusions, and using logical connectors and scientific phrases. The questionnaire is an adaptation of the instrument used by Vidal [41] to assess the level of written communication competence of engineering students at the same university.

To evaluate the level of knowledge achieved in the study, a student's t-test was used. Prior to its application, the Shapiro-Wilk test was conducted to test the normality of the data, and a $p$-value greater than 0.05 was obtained, indicating that the data followed a normal distribution. Cohen's d was used to calculate the effect size.

Throughout the semester, both the experimental and control groups covered the same course content in Writing and Communication. The difference was that the experimental group incorporated gamification into the course, using the gamified web platform *Call for Papers for Engineers*.

### 3.4. Description of the Gamified Platform

The gamified platform *Call for Papers for Engineers* consists of four sections: Home, Presentation, Writing in Engineering, and Writing Through Play. The latter is the gamified section, in turn it is mainly composed of a game titled *Call for Papers: The Game*, which

contains an interactive map to guide students in the fulfillment of the missions. It also contains a section of mini-games called 'GamiGames'. These mini-games encompass various activities, such as using references in APA and IEEE style, elements of the abstract, scientific phrases, connectors, crosswords, and more (Figure 1).

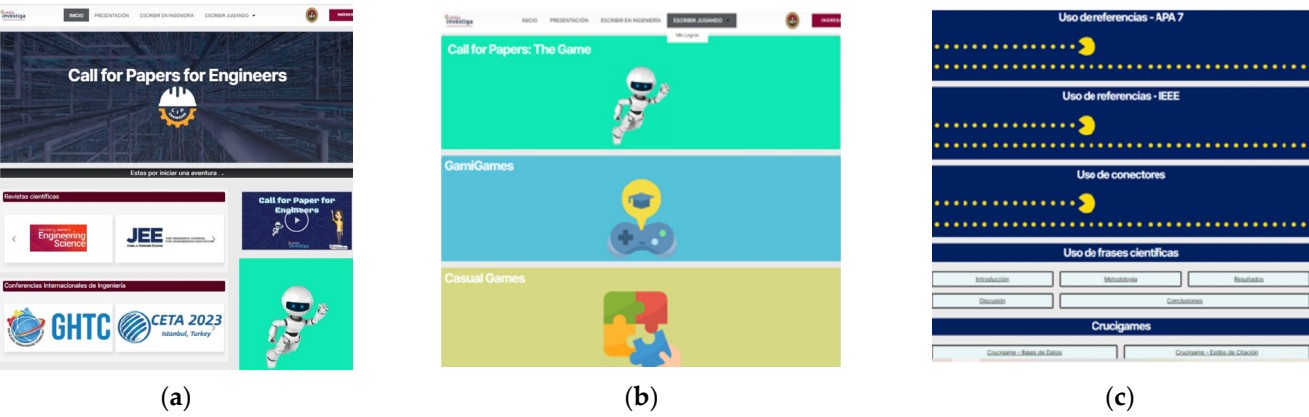

|       |       |       |
|:-----:|:-----:|:-----:|
| (**a**) | (**b**) | (**c**) |

**Figure 1.** The gamified platform *Call for Papers for Engineers*: (**a**) homepage; (**b**) playful writing; (**c**) gamigames.

### 3.5. Course Description and Course/Program Structure

The Writing and Communication course is organized into three units. The first unit covers topics related to reading, while the second and third units are specifically focused on scientific writing, covering its three phases: planning, text composition, and revision. The contents of the three units of the course were developed in both the experimental group and the control group, based on the same syllabus of the course. To complement the theoretical development, a series of quests were created for students to complete, and the primary support for this was provided through the gamified platform *Call for Papers for Engineers* (see Table 1).

**Table 1.** Course Description and Resource used.

| Missions to Be Completed. | Resources Used |
|---|---|
| Quest 1: Selection and reading of the specialty thesis | Thesis repository |
| Quest 2: Adaptation of the thesis into a scientific article format | Gamified platform *Call for Papers for Engineers* |
| Quest 3: Searching for articles in databases | Google Scholar, Scielo Scopus, Web of Science |
| Quest 4: Article planning | Gamified platform *Call for Papers for Engineers* |
| Quest 5: Article writing | Gamified platform *Call for Papers for Engineers* |
| Quest 6: Revision of the article | Gamified platform *Call for Papers for Engineers*, virtual classroom |
| Quest 7: Dissemination of the article | YouTube and Facebook platforms |

Likewise, a series of gamified activities called 'GamiChallenges' were created, which were contained within the *Call for Papers for Engineers* platform (see Table 2).

### 3.6. Validation of the Gamified Platform through Expert Judgment

To validate the gamified platform through expert judgment, the Questionnaire for Mobile Application Design by Jauregui-Velarde et al. [42] was adapted. The original questionnaire consists of 20 questions and five criteria: design, usability, functionality, security, and availability. For the purposes of this study, four criteria have been considered: design, usability, content, and instructional quality, with a total of 12 questions (see Table 3).

The evaluation questions required the judges to score on a scale defined by three levels: low, medium, and high. The three judges who participated in the evaluation have

experience in the fields of communication, software engineering, and higher education. Furthermore, they have an international background and hold doctoral degrees.

Descriptive statistics were used for the analysis of the judges' responses, and the mean and standard deviation were calculated for each of the questions, following the scoring system used by Jauregui-Velarde et al. [42]. A low level ranged from 0 to 1, a medium level from 1.1 to 2, and a high level from 2.1 to 3. As shown in Table 2, all 12 questions received assessments at the High level.

**Table 2.** Description of gamified activities.

| Gamified Activities | Description of Activity |
| --- | --- |
| GamiChallenge 1: Reading articles | Students read scientific articles related to the area of engineering. |
| GamiChallenge 2: Elements of the abstract | Students rearranged the elements of the abstract: background, methodology, results, and conclusion. |
| GamiChallenge 3: Relationship between abstract and keywords | Students should select keywords from the abstracts of the articles. |
| GamiChallenge 4: Using APA7 and IEEE references | Students should check for correctness of references in APA7 or IEEE styles. |
| GamiChallenge 5: Using connectors | Students should identify the different types of connectors. |
| GamiChallenge 6: Use of scientific phrases | Students should identify the most common scientific phrases for each part of the article. |
| GamiChallenge 7: CruciGame1 and CruciGame2 | Students should solve the proposed crossword puzzles. |
| GamiChallenge 8: Call for papers: The Game | Students should play the game *Call for Papers: The Game.* |

**Table 3.** Evaluation questions.

| Criteria | Design |
| --- | --- |
| P1 | Does the platform have a simple interface? |
| P2 | Does the platform present a pleasant visual environment? |
| P3 | Is the information on the platform well organized? |
| Criteria | Usability |
| P4 | Is the platform intuitive and easy to use? |
| P5 | Does the platform offer easy navigation? |
| P6 | Is the information on the platform user-friendly? |
| Criteria | Content |
| P7 | Is the content of the platform appropriate for teaching scientific writing in engineering? |
| P8 | Does the platform present valuable content for teaching engineering science writing? |
| P9 | Are the contents of the platform motivating for students? |
| Criteria | Instructional quality |
| P10 | Are the contents of the platform clear? |
| P11 | Do the exercises proposed on the platform facilitate the learning of scientific writing? |
| P12 | Does the use of the platform in general promote the learning of scientific writing in engineering? |

Regarding the mean scores in the Content criterion, the experts awarded the highest scores to questions 7 and 8, which refer to the appropriateness and value of the content for teaching scientific writing in engineering. Similarly, question 10 in the Instructional Quality criterion obtained the highest mean score (see Table 4).

### 3.7. Participants in the Experimental Group for the Use of the Gamified Platform

The satisfaction questionnaire was administered to 32 participants, with 25 (78.12%) being male and 7 (21.87%) being female. Among them, 13 (40.63%) were aged between 16 to 18 years, 16 (50%) were between 19 to 21 years, and 3 (9.37%) were 22 years or older. The participants who responded to the questionnaire were those who were part of the experimental group consisting of 33 participants (the first part of this study); one student

did not respond to this questionnaire. Participation was voluntary through a Google Form, and their informed consent was obtained.

**Table 4.** Result of the expert evaluation.

| Criteria | Question | Mean | DS | Nivel |
|---|---|---|---|---|
| Design | P1 | 2.33 | 0.577 | High |
| | P2 | 2.67 | 0.577 | High |
| | P3 | 2.67 | 0.577 | High |
| Usability | P4 | 2.33 | 0.577 | High |
| | P5 | 2.67 | 0.577 | High |
| | P6 | 2.67 | 0.577 | High |
| Contents | P7 | 3.00 | 0.000 | High |
| | P8 | 3.00 | 0.000 | High |
| | P9 | 2.67 | 0.577 | High |
| Instructional quality | P10 | 3.00 | 0.000 | High |
| | P11 | 2.67 | 0.577 | High |
| | P12 | 2.67 | 0.577 | High |

*3.8. Instrument for Satisfaction with the Use of the Gamified Platform*

To assess the level of satisfaction with the use of the gamified platform by the students in the experimental group, the Computer System Usability Questionnaire (CSUQ) version 3, adapted by Hedlefs et al. [43] for the Spanish context, was employed. The CSUQ is a questionnaire designed to evaluate overall user satisfaction. It consists of 16 items and has construct validity with three factors: system quality (items 7 to 12), information quality (items 13 to 16, 1, 3, and 4), and interface quality (items 2, 5, and 6).

In the overall 16-item questionnaire, a Cronbach's Alpha coefficient of 0.965 was obtained, which is similar to what was found by Hedlefs et al. [43]. In this research, participants from the experimental group were asked to evaluate the web-based gamified platform *Call for Papers for Engineers*, which was created to promote scientific writing.

Two open-ended questions were added to the questionnaire: "Do you consider that the gamified platform *Call for Papers for Engineers* can help you improve your scientific writing?" and "What suggestions would you give to enhance the gamified platform?".

## 4. Results

Table 3 displays the difference in means obtained in the experimental and control groups in the pre-test and post-test. In the pre-test, no differences were found between groups, whereas in the post-test, statistically significant differences were observed. The experimental group achieved a higher mean (15.82) in the post-test with a large effect size (d = 0.87). In the control group, no significant differences were found in the pre-test and post-test. These results highlight the level of knowledge regarding scientific article writing that the students in the experimental group reached using the web-based gamified platform *Call for Papers for Engineers* (see Table 5).

**Table 5.** Results of the knowledge test for the experimental group and the control group.

| Knowledge | Experimental Group | | Control Group | | t (58) | *p* | Cohen's d |
|---|---|---|---|---|---|---|---|
| | M | DS | M | DS | | | |
| Pre-test | 12 | 3.2 | 13.11 | 2.79 | −1.416 | 0.162 | |
| Post-test | 15.82 | 2.56 | 13.26 | 3.28 | 3.385 | 0.001 | 0.87 |

Note: The mean values for each analysis correspond to n = 33 for the experimental group and n = 27 for the control group.

No statistically significant differences were found based on gender or age. However, from a qualitative analysis of the knowledge test, women achieved a higher mean (16.86)

compared to the mean obtained by men (15.54), and participants aged 22 and older achieved the highest mean (17.33), followed by the 16 to 18 years age group (16.15), and the 19 to 21 years age group (15.00).

These results reveal that the course materials and the use of the web-based gamified platform *Call for Papers for Engineers* have been effective teaching procedures because students in the experimental group achieved higher performance after using the platform, unlike the control group, which did not show significant improvement compared to the initial assessment (pre-test).

Regarding the overall satisfaction measured by CSUQ (Computer System Usability Questionnaire), as shown in Figure 2, the means of the 16 items range from 5.28 to 6.16. In general, participants express satisfaction with the use of the platform. When examining the individual items, seven items scored above 6: Items 11 and 12 (quality of information) pertain to the information contained on the website. Items 16 and 1 (interface quality) are related to overall satisfaction with the website. Item 5 (system quality), which has the highest mean (6.16), pertains to the ease of learning how to use the website. In the dimension quality of information, the item with the lowest satisfaction was the item 7 "The website displays error messages that clearly tell me how to resolve the issues" (5.28).

Statistically significant differences by gender or age were not found; however, from a qualitative analysis of the satisfaction questionnaire, women achieved a higher mean score (6.16) compared to the mean obtained by men (5.79). Participants aged 16 to 18 years achieved the highest mean (5.98), followed by the group aged 19 to 21 years (5.85), and the group aged 22 years and older (5.50).

In relation to the mean scores obtained in the three factors, the scores are quite similar: 5.80 for quality of information, 5.89 for interface quality, and 5.96 for system quality. In this study, participants responded to the questionnaire after using the interface for the first time during the course, both in a face-to-face classroom setting and remotely. It's possible that issues with internet connectivity may have influenced the satisfaction reported by participants, as all three factors show scores below the highest possible mean of 7, which represents complete agreement with the web-based gamified platform *Call for Papers for Engineers* (see Figure 3).

In relation to the first open-ended question posed in the satisfaction survey, participants responded that the platform helps them improve their scientific writing because it contains examples based on real articles, provides a structured approach to writing, is a dynamic and enjoyable teaching method, offers easy and detailed instructions, imparts theoretical knowledge of writing, promotes self-directed learning, and provides vital tips for writing, among other responses.

Regarding the second question, which asked for suggestions to improve the gamified platform, the most relevant responses were focused on improving connectivity, enhancing the interface, upgrading servers, and addressing technical issues, as the platform sometimes experienced slowdowns. In a similar vein, other responses were more specific, such as "improving error messages in case of mistakes". These responses can partially explain the lower score obtained in the information quality factor (5.80) (see Figure 3), as participants suggested improvements in design to make the platform more attractive and correcting repeated questions that do not award points, among other suggestions.

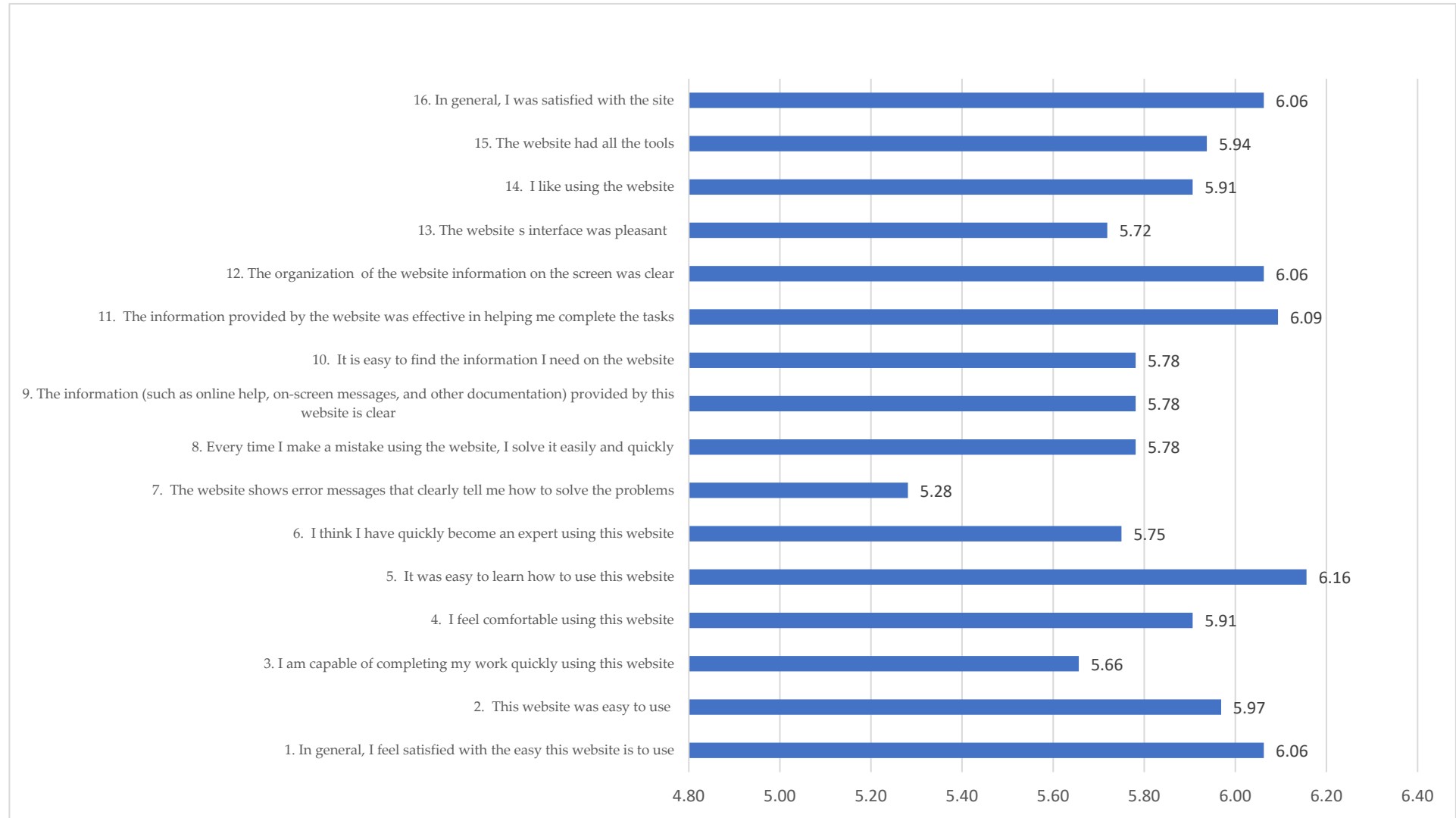

**Figure 2.** Mean of the satisfaction questionnaire of the web-based gamified platform *Call for Papers for Engineers*.

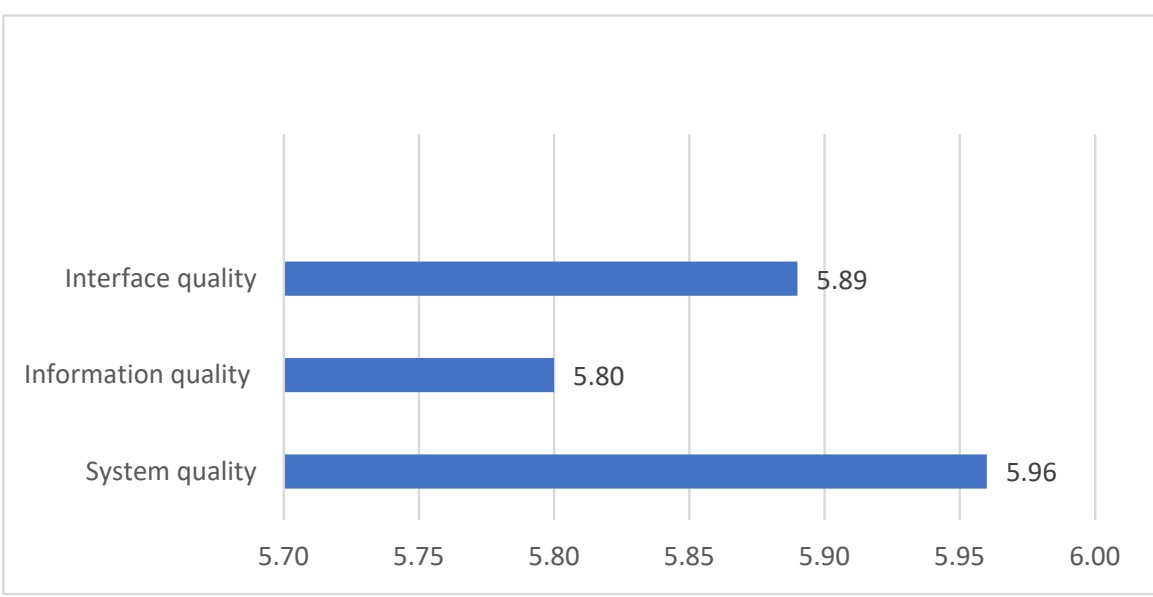

**Figure 3.** Mean of the factors in the satisfaction questionnaire of the web-based gamified platform *Call for Papers for Engineers*.

## 5. Discussion and Conclusions

This study primarily aimed to determine the potential of using a gamified platform in the development of scientific writing skills in engineering students. The results obtained are positive as they demonstrate the effectiveness of gamification in the "Writing and Communication" course through the utilization of the web-based gamified platform *Call for Papers for Engineers*. It was observed that students in the experimental group performed better after using the gamified platform, whereas students in the control group did not surpass their initial assessment results.

While there are gamified platforms in higher education that serve various purposes, such as κPAX, a gamified platform used for designing serious games in engineering [44], or EDUMAT, a gamified web tool for teaching mathematical operations [45], until now, there has not been a gamified platform designed specifically for scientific writing. In this regard, the web-based gamified platform *Call for Papers for Engineers* fills an existing gap in the field of scientific writing for engineering students. In other educational levels, such as primary and secondary education, some platforms with games have been proposed to promote STEM education [46].

Scientific writing in engineering requires certain considerations, as Li [33] refers to. Longo's work [47] asserts that to write effective documentation, an engineer must understand some pragmatic considerations. This means that their field of work operates within a specific environment and requires documentation with predefined characteristics. Therefore, it is important for engineers to become familiar with the documents they need to prepare and understand how those documents should appear.

The experience gained in this study has shown that the proposed web-based gamified platform plays an important role in helping engineering students improve their scientific writing. It achieves this by providing examples based on real articles and offering a structured approach to writing.

Similarly, Longo [47] emphasizes that writing can be approached pragmatically by observing forms, arguments, and words that align with the requirements of the engineer's context. This approach enables the creation of documents that "resemble" what is expected within the context. The proposed gamified platform, by using examples from real articles, provides this opportunity to students.

The root causes of the low proficiency in writing among engineering students can be complex and challenging to unravel. However, the reality is that many engineering

students lack the opportunity to develop or practice disciplinary writing in the subjects they study [48]. In a similar vein, previous studies [37,38,49,50] show that most interventions cover various strategies for incorporating writing reinforcement techniques within other courses. They provide models, examples, and structured writing exercises centered around authentic tasks. Unfortunately, most of these interventions have not achieved long-term continuity. It appears nearly impossible to permanently integrate writing into the engineering curriculum so that it is considered an integral element of becoming an engineer. This suggests that writing practices are not viewed as either developmental or intrinsic to the engineering curriculum [48].

Having a web-based gamified platform provides students with an opportunity to learn the key guidelines of the scientific writing process in a playful manner, with simple examples. Moreover, the platform is independent of a specific course and is a tool that students can revisit at any time for concepts, advice, or models to follow.

Furthermore, the participants' opinions about the motivating learning experience offered by the gamified platform align with the findings of Bybee [15], who asserts that educational video games enable the application of acquired knowledge, increase student motivation, and generate new didactic proposals that make learning more enjoyable.

The gamified platform required the creation of instructional materials that follow the logic of online games. Games were developed to reinforce the learning of the scientific writing process. These games focused on enhancing reading and writing skills related to engineering articles, recognizing the elements of scientific articles, and using references correctly, among other aspects. Additionally, specialized materials such as videos and infographics were created, and external free resources were linked. This approach aligns with the proposal of Dichev and Dicheva [8], who highlight that the incorporation of elements and mechanics from video games into education can reduce the lack of motivation while simultaneously promoting the development of higher-order cognitive skills and processes. The results obtained validate the impact of using the gamified platform.

One of the limitations of this study is that not all students who used the platform had adequate internet connectivity, which influenced their level of satisfaction with the platform. Another limitation relates to certain technical issues that arose, such as server failures, which made it challenging to extend the experience for a longer duration. In future work, the gamified platform will be applied to a larger sample with participants from other professional engineering schools.

In conclusion, the proposal of the web-based gamified platform *Call for Papers for Engineers* represents a contribution to students in the field of engineering because it allows them to enter the realm of scientific writing. In future work, there will be a greater implementation of the platform with educational content and games aimed at motivating students to further explore the world of science and technology.

**Author Contributions:** Conceptualization, R.N.-P.; methodology, R.N.-P. and A.B.-P.; software, R.N.-P., E.V. and E.C.-G.; investigation, O.T.-G.; formal analysis, A.B.-P.; writing—original draft preparation, E.V. and O.T.-G.; writing—review and editing, E.C.-G. and I.A.; supervision, I.A. All authors have read and agreed to the published version of the manuscript.

**Funding:** This research was funded by Universidad Nacional de San Agustín de Arequipa. It is part of the project "Transmedia Gamification and Video Games to promote scientific writing in Engineering students", under Contract No. IBA-IB-38-2020-UNSA.

**Institutional Review Board Statement:** IRB approval is waived due to the study required only information on the use of the gamified platform to improve their writing skills. In addition, the participation of the students was voluntary and informed and did not transgress any ethical principles governing scientific research. The confidentiality and protection of personal data of the participants was also respected.

**Informed Consent Statement:** Informed consent was obtained from all subjects involved in the study.

**Data Availability Statement:** The data presented in this study are available on request from the corresponding author. The data are not publicly available due to ethical restrictions.

**Acknowledgments:** To the Universidad Nacional de San Agustín de Arequipa and CiTeSoft for the support provided to the project.

**Conflicts of Interest:** The authors declare no conflict of interest. The funders had no role in the design of the study; in the collection, analyses, or interpretation of data; in the writing of the manuscript; or in the decision to publish the results.

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
