# Peer review of "Use of a Gamified Platform to Improve Scientific Writing in Engineering Students"

_education, doi:10.3390/educsci13121164_

Round 1

Reviewer 1 Report

Comments and Suggestions for Authors

The paper titled "Use of a gamified platform to improve scientific writing in engineering students" explores the potential of gamification in enhancing scientific writing skills among engineering students at a Peruvian university. The study employs a quantitative approach with a quasi-experimental design, comparing an experimental group that used the gamified web platform, "Call for Papers for Engineers," to a control group. The abstract succinctly presents the key aspects of the research, but there are several points to consider.

The research field addressed in this paper is valid, and the references made to the importance of good writing skills for engineering students support the significance of the study. Understanding the context and need for enhancing scientific writing skills is crucial to appreciating the relevance of the authors' work.

The methodological approach used to assess the impact of the gamified platform on scientific writing skills is scientifically sound. The use of a quasi-experimental design with pre- and post-test measurements helps in understanding the effectiveness of the gamified platform. However, the abstract does not provide sufficient details about the game and its design. Describing the game and its features in more detail would be essential for readers to evaluate the impact accurately. Information on the gamified platform's structure, elements, and how it facilitated the improvement of writing skills is vital for disseminating knowledge effectively.

The conclusion of the study is presented clearly and is easy to understand. However, the claim that the used game "represents a significant contribution" might be exaggerated, especially if the sample size (n) was relatively small. To strengthen this assertion, the authors should consider conducting further research with a larger sample size. This would provide more robust evidence of the game's impact on scientific writing skills.

In summary, the paper is a valuable contribution to the field of higher education for engineering students. It addresses an important issue and employs a valid research methodology. To enhance the paper, the authors should provide a more detailed description of the gamified platform and its design. Additionally, further research with a larger sample size could support the claim of a "significant contribution." Finally, minor proofreading to address spelling and paragraph structure would improve the overall presentation of the paper.

Comments on the Quality of English Language

minor proofreading to address spelling and paragraph structure

Reviewer 2 Report

Comments and Suggestions for Authors

Strengths of the paper

The paper discusses the potential benefits of using gamified platform in the classroom, as well as some of the potential drawbacks. The article is likely to be of interest to the readership of the Education Sciences journal, including educators, researchers, and parents. The discussion of the potential benefits and drawbacks of using gamified platform in education is relevant and interesting, as such the findings may be insightful to educators who are considering using it in their classrooms.

Points need further improvement.

·      METHOD. I have serious concerns regarding the design of the survey and the sample on which the analysis is based. First, there is no information if the sample analyzed by the Authors is representative. The sample size and its representativeness mean that the soundness of the research conclusions is questionable. This section must be improved.

·      RESULTS. I consider that figure 3 is not the most appropriate to show these results.

I hope my suggestions will help you.

Reviewer 3 Report

Comments and Suggestions for Authors

Thank you for the opportunity to read this. The research titled "Use of a gamified platform to improve scientific writing in engineering students" offers an intriguing exploration into the realm of educational innovation. The study, conducted at a Peruvian university, delves into the potential of leveraging gamification to bolster the scientific writing capabilities of engineering students. Through the design and implementation of a gamified web platform named "Call for Papers for Engineers," the researchers adopted a quantitative methodological approach, juxtaposing the outcomes from an experimental group against a control group. They suggest a positive correlation between the gamified platform's use and enhanced student performance. I have mixed feelings about this research. Because the authors predominantly highlight the positive effects of gamification on enhancing the academic writing skills of engineering students, I would appreciate further clarification to fully gauge the significance of this study. I have some questions that, if addressed, might help the readers in understanding the implications and merits of this research, and to ensure its suitability for publication.

 1. Authors suggest that gamification universally enhances communicative competencies (“Studies have shown that gamification enhances communicative competencies in all target language skills, opening up new opportunities for innovation in teaching processes”). However, the effectiveness of gamification might vary based on individual preferences, learning styles, and the specific design and implementation of the gamified approach. Not all gamified platforms or methods might be effective for all learners. What’s your take on this?

 2. I agree that gamification can make the learning process more engaging, but there is a question of depth in my view. Does gamifying writing truly foster a deep understanding and appreciation for the intricacies of writing in engineering context, or does it simply make surface-level learning more fun?

 3. Also gamification may offer immediate engagement and motivation, but what about long-term motivation and interest? Once the novelty of the game elements wears off, will students still be motivated to improve their writing, or will they only be motivated when presented with game-like elements?  

 4. In my experience, academic writing is very intricate, requiring critical thinking, extensive research, and the ability to convey complex ideas clearly. Can game mechanics truly capture and teach the depth and breadth of these skills, or do they risk oversimplifying the process?

I couldn’t find answers to these questions (and extensively include in the introduction and literature review), and without them this whole research seems to be scratching the surface without providing an in-depth understanding of the effects of gamification on academic writing. Again, the authors assume that all students, being "digital natives," will naturally gravitate towards and benefit from gamified learning. However, individual preferences can vary widely, and not all students might find gamified learning appealing or effective.

5. I am also wondering: As the academic writing landscape continues to evolve, one cannot also help but ponder the tangible benefits of such an approach, especially when juxtaposed against the growing capabilities of generative AI. As AI systems become increasingly adept at assisting, and in some cases, autonomously producing academic content, the allure of gamified platforms might wane. While the study's creative approach is worth mentioning, its practical application might be challenged in an era where AI-driven tools could provide students with more direct and potentially more engaging means to hone their academic writing skills.

Some other comments are below:

 a. The introduction and Literature review should be separated.

b. Also the text in Intro section often revisits the same topics or concepts without adding substantial new information or perspectives. For instance, the benefits of gamification in education, its use in enhancing motivation, and the challenges faced by students in academic writing are repeated in multiple places.

c. While the section starts by discussing various methodologies to improve writing competence, it delves deeply into gamification without adequately addressing the other methodologies mentioned (It would be beneficial to clearly segregate the introduction, the general premise of gamification, its specific application in academic writing, and then its role in scientific writing within engineering.).

d. Methods/data: There is no mention of whether the questionnaire used to assess knowledge has been validated previously or piloted for this specific context. The reliability and validity of the instrument are critical for the study's findings.

e. Authors mentioned that both the experimental and control groups covered the same course content, but to me it not clear if the delivery method was identical (barring the gamification for the experimental group)??.

Comments on the Quality of English Language

Minor editing needed 

Round 2

Reviewer 3 Report

Comments and Suggestions for Authors

The authors refined the paper and the revised version is satisfactory.